# Use of Licorice (*Glycyrrhiza glabra*) Herb as a Feed Additive in Poultry: Current Knowledge and Prospects

**DOI:** 10.3390/ani9080536

**Published:** 2019-08-07

**Authors:** Mahmoud Alagawany, Shaaban S. Elnesr, Mayada R. Farag, Mohamed E. Abd El-Hack, Asmaa F. Khafaga, Ayman E. Taha, Ruchi Tiwari, Mohd. Iqbal Yatoo, Prakash Bhatt, Gopi Marappan, Kuldeep Dhama

**Affiliations:** 1Department of Poultry, Faculty of Agriculture, Zagazig University, Zagazig 44511, Egypt; 2Department of Poultry Production, Faculty of Agriculture, Fayoum University, Fayoum 63514, Egypt; 3Forensic Medicine and Toxicology Department, Faculty of Veterinary Medicine, Zagazig University, Zagazig 44511, Egypt; 4Department of Pathology, Faculty of Veterinary Medicine, Alexandria University, Edfina 22758, Egypt; 5Department of Animal Husbandry and Animal Wealth Development, Faculty of Veterinary Medicine, Alexandria University, Edfina 22758, Egypt; 6Department of Veterinary Microbiology and Immunology, College of Veterinary Sciences, UP Pandit Deen Dayal Upadhayay Pashu Chikitsa Vigyan Vishwavidyalay Evum Go-Anusandhan Sansthan (DUVASU), Mathura-281001, Uttar Pradesh, India; 7Division of Veterinary Clinical Complex, Faculty of Veterinary Sciences and Animal Husbandry, Jammu and Kashmir, Srinagar 190006, India; 8Teaching Veterinary Clinical Complex, College of Veterinary and Animal Sciences, Govind Ballabh Pant University of Agriculture and Technology, Pantnagar-263145 (Udham Singh Nagar), Uttarakhand, India; 9Division of Avian Physiology and Reproduction, ICAR-Central Avian Research Institute, Izatnagar, Bareilly-243 122, Uttar Pradesh, India; 10Division of Pathology, ICAR-Indian Veterinary Research Institute, Izatnagar, Bareilly-243 122, Uttar Pradesh, India

**Keywords:** licorice, *Glycyrrhiza glabra*, beneficial effects, pharmaceutical, poultry, health

## Abstract

**Simple Summary:**

The present review updates the current knowledge about the beneficial effect of licorice supplementation in poultry diets, particularly its positive effect on the treatment of high-prevalence diseases of the immune system, liver, and lungs.

**Abstract:**

Supplementation of livestock and poultry diets with herbal plants containing bioactive components have shown promising reports as natural feed supplements. These additives are able to promote growth performance and improve feed efficiency, nutrient digestion, antioxidant status, immunological indices, and poultry health. Several studies have used complex herbal formulas with the partial inclusion of licorice. However, the individual use of licorice has been rarely reported. The major problem of the poultry industry is the epidemiological diseases, mainly confined to the respiratory, digestive, and immune systems. Licorice has certain bioactive components such as flavonoids and glycyrrhizin. The roots of this herb contain 1 to 9% glycyrrhizin, which has many pharmacological properties such as antioxidant, antiviral, anti-infective and anti-inflammatory properties. Licorice extracts (LE) have a positive effect on the treatment of high-prevalence diseases such as the immune system, liver, and lung diseases. Studies showed that adding LE to drinking water (0.1, 0.2, or 0.3 g/L) reduced serum total cholesterol (*p* < 0.05) of broiler chickens. Moreover, LE supplementation in poultry diets plays a significant role in their productive performance by enhancing organ development and stimulating digestion and appetite. Along with its growth-promoting effects, licorice has detoxifying, antioxidant, antimicrobial, anti-inflammatory, and other health benefits in poultry. This review describes the beneficial applications and recent aspects of the *Glycyrrhiza glabra* (licorice) herb, including its chemical composition and role in safeguarding poultry health.

## 1. Introduction

Medicinal plants have gained great popularity for their several beneficial applications in animals, poultry, and humans [1,2]. Nowadays, the addition of feed additives and nutritional supplements, including prebiotics, plant extracts, and probiotics, in the diets of birds are gaining wide attention owing to their multiple beneficial applications while enhancing growth performances and production as well as safeguarding the health of poultry [1,3,4,5]. This review focuses on the use of herb licorice (*Glycyrrhiza glabra*) as a feed additive in poultry, a popular traditional medicinal plant that belongs to the legume family Fabaceae [6]. It is broadly used in the medicine sector, as a flavouring and food preservative agent and also for commercial purposes [7]. It is derived from the sweet root of various species of *Glycyrrhiza*; however, the cultivation and harvesting practices modify the composition of various biologically important components of the *Glycyrrhiza* plant [8]. Phytochemical analysis showed that the major fraction of licorice extract (LE) consisted of triterpene saponins (e.g., glycyrrhizin, glycyrrhetinic acid, and licorice acid), flavonoids (e.g., liquiritin, isoflavonoids, and formononetin), sugars, starch, amino acids, ascorbic acid, tannins, choline, coumarins, phytosterols, and some other bitter principles [7,9]. Importantly, numerous pharmacological effects have been described for LE and its isolated active principles in humans and animals [7]. Licorice represents a replacement candidate reported to be useful for its multiple beneficial health effects including immunomodulatory, antimicrobial, antioxidative, anti-inflammatory, antidiabetic, hepatoprotextive, antiviral, anti-infective, and radical-scavenging activities [7,8,10]. This review describes the beneficial applications and recent aspects of the licorice herb, including its chemical composition, health benefits, and useful applications for nutritionists, physiologists, scientists, pharmacists, veterinarians, pharmaceutical industries, and poultry breeders. Therefore, we can safely assess and get a new vision for further research on licorice benefits in poultry nutrition and its effects on the growth and productive performance and immune and antioxidant status of poultry.

## 2. Chemical Composition and Structure

Licorice is also known as Radix Glycyrrhizae or Liquiritiae Radix. It is the root of *Glycyrrhiza uralensis* Fisch. ex DC., *G. glabra* L. or *G. inflata* Bat., Leguminosae [10,11]. The roots of *G. glabra* (Figure 1) are widely used in preparing several pharmaceutical preparations. Phytochemical analysis of licorice root extract exhibited that it contained flavonoids (isoflavonoids, formononetin, and liquiritin), saponin triterpenes (liquirtic acid and glycyrrhizin), and other components such as sugars, coumarins, amino acids, starch, tannins, phytosterols, choline, and vitamins (e.g., ascorbic acid) [7,9,12]. Previous reports have shown that more than 20 triterpenoids and 300 flavonoids have been procured from licorice [13]. Glycyrrhizin constitutes up to 25% of the licorice root extract [14]. Glycyrrhizin consists of glucuronic acid (two molecules) and glycyrrhetinic acid (one molecule) [15]. Badr et al. [16] analyzed the raw form of licorice chemically and summarized its contents as follows: carbohydrate (47.11%), fiber (24.48%), protein (9.15%), silica (3.56%) and low fat content (0.53%). Moreover, the ash and moisture content values of the licorice root were found to be 7.70 and 6.80%, respectively. Additionally, the same authors reported that the calcium and phosphorus content values of the raw LE were 1720 and 78 mg/100 g, respectively, and the major components of amino acids that were found in LE were proline (1.02%), aspartic (0.88%), alanine (0.51%) and glutamic acid (0.50%). 

The licorice root color is yellow because of its flavonoid components such as hispaglabridins and glabridin [17]. Additionally, the dried aqueous extracts of licorice contain approximately 4–25% glycyrrhizinic acid [18]. The main active ingredients of licorice are liquiritin, isoliquiritigenin, liquiritigenin, and glycyrrhetinic acid, glycyrrhiza polysaccharide, and this herb is rich in flavonoids and syringic, abscisic, trans-ferrulic, 2,5-dihydroxy benzoic, abscisic, and salicyclic acids [7,9,19]. Pharmacological activities are contributed to by glycyrrhizin, 18β-glycyrrhetinic acid, glabrin A and B, and isoflavones of *Glycyrrhiza glabra* Linn [7]. 

## 3. Beneficial Role of Licorice

In ancient times, *G. glabra* was used as a medicine and flavouring herb. It is a soothing herb that enhances various body functions, protects the liver and is used in various conditions, such as mouth ulcers and arthritis, and as a potent anti-inflammatory, immunomodulatory, hepatoprotective, detoxifying, anti-cancer, anti-aging, antioxidant, antimicrobial, with growth promoting effects [6,8,9]. The licorice herb has several biological activities and health promoting effects, as are discussed in the following sections. 

### 3.1. Antioxidant and Anti-Inflammatory Activities

Previous phytochemical analyses have revealed that the bioactive components of the licorice root include flavonoids (isoflavonoids and liquiritin), glycyrrhizic acid, liquiritigenin, triterpenes (glycyrrhizin), and saponins, which have anti-inflammatory and antioxidant properties [20,21,22,23]. Various modes of action with regards to antioxidant and anti-inflammatory properties of licorice can be narrated as: licorice extracts inhibits the lipid peroxidation of mitochondria, decreases the oxidative rate and reactive substance formation of thiobarbituric acid; protects from scavenging free radicals; stimulates antioxidant enzyme activities; inhibits the activity of phospholipase A2 that acts as a critical enzyme in various inflammatory processes; Licochalcone inhibits lipopolysaccharide- induced inflammatory responses; Lico A derived from the licorice root inhibits the lipopolysaccharide-induced inflammatory responses in a dose-dependent manner by suppressing the activation of NF-κB and p38/ERK MAPK signaling; licochalcone A prevents cellular oxidation; licorice flavonoids renders a pro-inflammatory action; flavonoids might target the NF-κB signaling pathway to prevent the secretion of inflammatory cytokines; glycyrrhizic acid, liquiritigenin, and liquiritin can reduce the expression levels of pro-inflammatory cytokines (IL-1β, IL-6, and TNF-α) in the liver, and block the generation of several inflammatory mediators created by activated macrophages; glycyrrhizic acid directly inhibits prostaglandin E2 formation and cyclooxygenase activity and indirectly inhibits platelet aggregation and inflammatory factors [8,14,23,24,25]. Glycyrrhetinic acid might lead to the delayed secretion of cortisol with subsequent high levels of oxidation that resulted in increased heart weight in hens [26]. The proven and potent anti-inflammatory and antioxidant properties of the licorice herb need to be studied in poultry for the safeguarding of the heath of birds in poultry production.

### 3.2. Immunomodulatory and Antiviral Effects

Herbs have proven potent immunomodulatory and antiviral activities [2,25,27]. The extracts of licorice have a positive effect on the immune system of poultry. It can be used to optimize their immune response and improve the productive performance. The dietary supplementation of 0.1% LE improved the humoral immunity in broilers by inducing antibody titres against non-specific and specific antigens. An experiment was conducted to assess the effect of the supplementation of licorice root extracts on the immune profile of 54 commercial broiler chicks. The serum biochemical parameters, such as serum total protein, albumin, globulin, and albumin/globulin ratio, were estimated from three groups of chicks. Chicks were categorized into three groups; one in which the birds were provided with 1% *G. glabra* crude extract powder, the second one in which 0.1% *G. glabra* extract powder was given, and the third one in which no *G. glabra* extract was given. Humoral immunity was assessed by measuring the hemagglutination inhibition (HI) titre against the Ranikhet disease virus (LaSota) and the hemagglutination antibody (HA) titre against sheep red blood cells (SRBC) antigens, while the cell-mediated immune response was measured by estimating the total and differential white blood cell (WBCs) counts. The results of the study revealed that the chicks supplemented with 0.1% licorice extract powder showed considerable improvement in their immune responses [28]. Also, natural feed supplements are used as immunity enhancers because it increases WBC counts and ultimately boosts interferon levels [29]. Furthermore, Dorhoi et al. [30] stated that the addition of LE (50 μg/mL) to the diet of laying hens had some beneficial effects on their cellular immunity. The glycyrrhiza polysaccharide has a sturdy immune action and is widely involved in some features of immune regulation [31]. Additionally, LE increased the phagocytic capacity of mononuclear cells and granulocytes of chicken [30]. 

Notably, the addition of licorice in broiler diets improved the weight of immune organs, such as the spleen or bursa, thereby promoting immune efficacy and the situation of livability and health [32]. Glycyrrhetinic acid has several favourable pharmacological properties, such as immunomodulation and production of interleukins [1,2,12] with subsequent production of antibodies, gamma interferon, and T-cells, which indicates its antiviral activity [33]. However, Hosseini et al. [34] reported that the supplementation of the broiler diet with LE (2.5 and 5 g/kg diet) had no effect on immune organ weights. 

The active components of licorice and its extracts have anti-inflammatory, immunomodulatory and antiviral functions, and thus can augment the immunity of poultry by modulating both humoral and cell-mediated immune responses, prevent viral diseases, and render supplementary treatment for viral diseases [35,36]. Omer et al. [35] reported that LE (60 mg/100 mL *Glycyrrhiza* extract) when used as a phytogenic feed additive exhibited antiviral activity against the Newcastle disease virus (NDV). Moreover, the broilers treated with glycyrrhizic acid (GRA) at the concentration of 60 µg /mL drinking water showed higher antibody titers against the ND virus as well as an enhanced cellular immune response, as indicated by an increase in blood lymphocyte and thrombocyte counts [36]. In an in vivo antiviral study, a dose of 300 µg/mL of *G. glabra* extract showed potent antiviral action against NDV. Survival rates were higher in embryonated egg groups inoculated with NDV and treated with extract, and no virus was recovered in allantoic fluids in such groups which indicated the effective control of the virus by the herbal extract [5]. Dziewulska et al. [37] stated that dietary LE (10% extract) supplementation inhibited paramyxovirus type 1 (PPMV-1) replication in pigeons, and the copy number of viral RNA in some organs, such as the kidney and liver, of the pigeons fed LE was lower compared with the control pigeons, suggesting that LE has antiviral effects. In an aqueous solution of LE when administered at a dose rate of 300 or 500 mg/kg body weight to pigeons inoculated with PPMV-1 for 7 days, the expression of the IFN-γ gene was found to be increased in all PPMV-1 inoculated and herbal extract treated pigeons. Expression of the CD3 gene was lowest at 7 dpi in treated birds. CD4 gene expression was higher in uninoculated and treated pigeons but was lower in extract treated pigeons inoculated with PPMV-1. The CD8 gene also showed a non-significant difference in inoculated and extract treated pigeons, and the percentage of IgM^+^ B cells was also not affected [38]. The immunomodulatory and antiviral effects of licorice observed in pigeons also reveals its potent application to be explored and applied in poultry and other avian species to counter viral diseases. However, Moradi et al. [39] reported that the antibody titres against Newcastle disease (ND) and avian influenza (AI) viruses, as well as liver and lymphoid organ (e.g., bursa of Fabricius, thymus and spleen) weights, were not affected by LE supplementation in broiler drinking water. Glycyrrhizin has been reported to act as an immune stimulant for ducklings, inhibits the cytopathic effect of duck hepatitis virus (DHV) in VERO cells, potentiates the production of higher antibody titer in a DHV vaccinated group, and demonstrates a pronounced lymphocytic proliferation response, indicating its antiviral effects [40]. Glycyrrhizin has also been reported to inhibit influenza A virus uptake into the cell, mediated by its interaction with the cell membrane, which, in turn, results in reduced endocytotic activity and decreased virus uptake [41]. Thus, several studies have confirmed the positive impact of licorice on the immune potential and anti-viral effects in poultry; however, further studies are recommended to optimize the inclusion levels of LE in poultry diets and to determine their possible physiological and protective effects, as well as economic value. 

Regarding the protective role of licorice against aflatoxicosis in broilers, Al–Daraji et al. [42] reported that the addition of licorice to the aflatoxin (AF)-contaminated diet (at the concentrations of 150, 300, or 450 mg licorice/kg of diet) significantly recovered the adverse effects of AF on most carcass traits.

Saponins from *G. glabra* in combination with antigens from *Eimeria tennela* have shown the ability to serve as immune-stimulating complexes (ISCOMs) and provides immunity against avian coccidiosis, caused by *E. tennela* [43]. They protected birds from the experimental challenge of *E. tennela* and additionally, the antibody titer (IgG and IgM) against a homologous challenge was found to be increased. Saponins have also been identified to act as effective delivery units for antigens for vaccine development, have shown no toxicity and provided stronger immunity [44]. Being natural constituents of plants, saponins of *G. glabra* are unlikely to have any side effects and are comparable to *Aesculus hippocastanum*, *Gipsophila paniculata* or Quil-A saponins [43,44]. More recently, saponins derived from *G. glabra* (Glabilox) have shown better adjuvant potential and the ability to be used as immunostimulatory complexes along with antigens for vaccine purposes. *G. glabra* derived saponins were not found to be toxic or hemolytic as compared to Quil-A saponins, and could produce stable immunostimulatory complexes, hence were preferable as safe and effective vaccine adjuvants [33]. Glabilox induced strong humoral and cellular immune response against H7N1 influenza virus antigens on subcutaneous inoculation and provided 100% protection against homologous infection in chickens [33]

### 3.3. Effect of Licorice on Some Blood Chemistry

Broiler chickens given drinking water supplemented with LE (0.1, 0.2, or 0.3 g/L) showed reduced serum glucose, LDL cholesterol, and total cholesterol levels (*p* < 0.05) as well as reduced gall bladder weight [45]. The inclusion of *G. glabra* extract (0.5%) in broiler diets induced an increase in serum globulin concentration, which, in turn, led to an improvement in the humoral immune status [46]. However, the broilers fed 0.5 and 1 g licorice/kg during their growing period showed an increase in the number of WBCs (*p* <0.05) compared to the control. Furthermore, dietary licorice supplementation (0.5, 1, and 2 g/kg) did not have significant effects on the lymphocyte (L), heterophil (H), and monocyte percentages, heterophil to lymphocyte (H/L) ratio, and proliferation of red blood cells [47]. Moreover, the heterophil and lymphocyte percentages and H/L ratio were not affected by LE supplementation (0.1, 0.2, and 0.3 mg/L) in drinking water [39]. The licorice root enclosed phytoestrogens that boosted the rate of erythrocyte sedimentation and decreased the number of erythrocytes [48]. Additionally, the injected LE stimulated cell cycle and activity in lymphocytes [49]. Furthermore, Sharifi et al. [50] clarified that the licorice root supplementation in broiler diets (2 mg/kg diet) reduced some serum components, such as triglycerides, cholesterol, and LDL, and increased the high-density lipoprotein (HDL) levels. In another study, Sedghi et al. [47] concluded that the concentrations of cholesterol and LDL significantly declined in the birds fed diets containing licorice (0.5, 1, and 2 g/kg) compared to the control. This might be attributed to the inhibition of lipid peroxidation and lipoxygenase and cyclooxygenase enzyme activities as well as reduction of LDL oxidation by licorice. The cholesterol-lowering effects of LE are attributed to the high secretion of cholesterol, bile acids, neutral sterols, and improvement in the content of hepatic bile acid. Besides, the active components of licorice (saponin) are able to reduce the levels of LDL-associated carotenoids, inhibit the formation of lipid peroxides, and enhancement of the rate of conversion of cholesterol to bile acids with subsequent hepatic clearance. However, the feeding of licorice (0.5, 1, and 2 g/kg) in the study of Sedghi et al. [47] did not have a significant influence on the concentrations of triglycerides, HDLs, VLDLs, and glucose in the blood serum of broilers. Al-Daraji [51] concluded that the high levels of LE (150 to 450 mg/L in water) augmented glucose concentrations in the serum of broiler chickens under heat stress. 

The dietary supplementation of LE increased the HDL concentration and HDL/LDL ratio in serum because of its high concentrations of flavonoids and ascorbic acid [45]. Moreover, the inclusion of 0.4% LE in the drinking water of broilers increased the plasma HDL levels, but reduced the level of alanine aminotransferase (ALT) (*p* < 0.05) [52]. However, Shahryar et al. [53] concluded that the serum blood parameters of the laying hens supplemented with different concentrations of licorice powder (0.5, 1.0, 1.5, and 2.0%) did not significantly (*p* > 0.05) vary compared to the control group. Thus, the presence of saponins and phytosteroids in LE could be essential for removing cholesterol and increasing the content of hepatic bile acid in animals fed LE diets. *G. glabra* produced lower abdominal fat percentage (*p* < 0.05) in broilers given 0.3 g/L of LE in drinking water [6]. Moreover, the supplementation of LE at levels 0.1, 0.2, and 0.3% in drinking water decreased the concentrations of serum LDL, total cholesterol, and glucose (*p* < 0.05) [45].

### 3.4. Effect of Licorice on Some Growth Parameters and Performance

Currently, it is well established that the growth and laying performances of poultry are usually improved via supplementation of feed additives or growth promoters, which have a positive influence on their general health state and growth performance [1,3]. The inclusion of 0.4% LE in the drinking water of broilers increased (*p* < 0.05) the feed intake at 21 and 42 days, but did not affect the body weight at different ages [51]. However, Jagadeeswaran and Selvasubramanian [28] found that the inclusion of 1% LE to the basal diet of the broiler chickens improved their body weight and FCR at 42 days of age in comparison to the control group. In Japanese quails [54] it was reported that the inclusion of 200 ppm of licorice root extract containing 1% probiotic supplement to the quail diet improved the amount of daily feed intake and body weight gain. Furthermore, LE had positive effects on the productive performance of heat-stressed broiler chickens [55,56]. *G. glabra* diet supplementation in poultry positively affected their growth performance by enhancing the development of their organs. Furthermore, the digestion and appetite improved in broilers fed diets supplemented with 2.5 g/kg *G. glabra*. Moreover, the inclusion of up to 0.5% *G. glabra* in poultry diets during the pullet growing period enhanced the performance of laying hens [6]. 

Concerning the use of glycyrrhizic acid (GRA), the broilers supplemented with GRA (60 µg/mL in water) had higher body weight gain (BG), final body weight, better FCR, and the lowest mortality rate compared to the non-treated controls [36]. The feed intake of the laying hens fed 0.5, 1.0, 1.5, and 2.0% of licorice powder added to the basal diet was not affected [52]. Simultaneously, Hosseini et al. [34] used 5 g licorice/kg broiler diet and found no significant effect (*p* > 0.05) on body weight, feed intake, FCR, livability, and production index. Additionally, Moradi et al. [39] concluded that the inclusion of 0.1, 0.2, and 0.3 mg LE/L drinking water for broiler chicks had no significant effect on their body weight, feed intake, and FCR compared to the control group. Moreover, Sedghi et al. [47] used 0.5, 1, and 2 g LE/kg broiler diet and reported no effect on broiler weight, feed intake, and FCR compared to the non-supplemented group. However, another study reported different results when the percentage of licorice was modified and given in combination. This study was performed to determine an improvement in the productive traits of 180 one-day-old Ross 308 broiler chicks fed diets supplemented with different concentrations of licorice and garlic mixture powders. It was concluded that the diet supplemented with a mixture of garlic and licorice (at 0.25, 0.50, and 1% concentrations) improved the productive performance of broiler birds [57]. Another study carried out on 480 one-day-old male broiler chicks (Ross 308) showed the beneficial effects of 1% LE on the growth performance, immune system, and blood parameters of broilers, when supplemented along with the extracts of other plants, such as German chamomile, yarrow, eucalyptus, Iranian caraway, and garlic, and one antibiotic virginiamycin [58]. One more study performed on 400 unsexed (Cobb 500) broiler chicks advocated that LE reduced the abdominal fat of chicks without illustrating any adverse effects on their immune status and performance of broiler when receiving LE in drinking water for 42 days [59]. 

Experiments were performed to determine the effect of diet supplementation with thyme, peppermint, green tea, and licorice in 245 one-day-old broiler chickens on enhancing their growth performance, serum lipid profile, immune response, and carcass characteristics. An aqueous blend of 400 g thyme extract, 300 g peppermint extract, 200 g green tea extract, and 100 g LE, which finally provided 1.4% essential oils (0.4% thyme oil/0.9% peppermint oil), 25% polyphenols, 15% catechins, and 0.5% glycyrrhizinic acid as active principles, was used. A total of seven experimental groups were set up: five groups supplied with 100, 200, 500, 1000, and 2000 ppm of the aqueous blend and one negative (no supplementation) and one positive (antibiotic: oxytetracycline) control group. The results depicted an overall increase in the performance of chicks at 200 and 1000 ppm levels as compared to the negative control and a significant increase in humoral immunity as compared to the positive control. These findings recommended the inclusion of the plant extract blend (at 200 ppm concentration) in poultry diet to support the broiler performance and immune status, in addition to its use as a growth promoter and an alternative to antibiotics [60]. 

A study performed on hundred 40-week-old laying hens showed that the diets supplemented with LE improved the production of functional eggs and modulated the productive performance of laying hens by lowering the egg cholesterol level, some plasma parameters (reduction in LDL and egg-yolk cholesterol level with an increase in the HDL level and total antioxidant capacity of plasma [61]. Scientific literature has witnessed the global impact of this herb on the performance, carcass traits, and meat quality of poultry, when their diet is supplemented with licorice (*G. glabra*) either in feed and/or in drinking water [6]. In broiler chickens, probiotic and licorice extract (500 ppm) increased body weight gain of broilers exposed to high stocking density [62]. Drinking water supplementation of LE has been suggested to be an alternative to in-feed antibiotic growth promoter in broiler chickens [45]. Body weight gain (BWG) was improved in birds reared at high stocking density by licorice extract (500 ppm) but not the feed conversion ratio (FCR) [62]. This was further improved by the addition of probiotics (200 ppm). 

An overview of the beneficial effects and modes of action of *Glycyrrhiza glabra* in poultry health and production is depicted in Figure 2.

## 4. Conclusions and Future Prospects

The extract of *G. glabra* might play an important role in the preparation of several pharmaceutical compounds for further use in the poultry industry. Licorice contains bioactive components, such as flavonoids and glycyrrhizin, which have pharmacological properties and medicinal applications. The licorice extract has been found to show immunogenic and antioxidant activities, which might improve the growth performance, feed efficiency, carcass traits, and blood biochemical indices of the poultry birds, and act as a potential solution for solving respiratory, digestive, and immune problems in poultry. The use of LE up to 0.4 g/L in the drinking water of poultry increased the feed intake, and improved the immune response and antioxidant parameters as well as lipid profile. The addition of LE at 50 μg/mL in the diet of laying hens has been found to produce some beneficial effects on their cellular immunity. A dose of 300 µg/mL of *G. glabra* extract showed potent antiviral action against NDV. Further studies need to be conducted to evaluate the beneficial effects of using the licorice herb as poultry feed additive, as well as to explore other properties of this medicinal plant that might enhance productivity and health in poultry. Efforts need to be made to enhance the delivery of this important herb in poultry by exploring the nanodelivery and in ovo delivery techniques, thereby efficiently enhancing production and safeguarding the health of birds in a better way.

## Figures and Tables

**Figure 1 animals-09-00536-f001:**
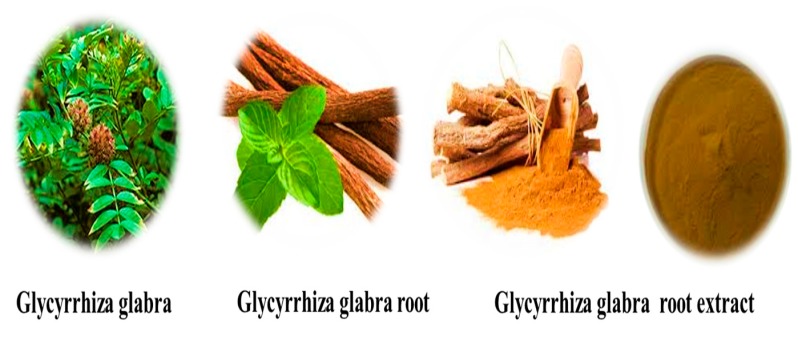
Pictorial representation of the *Glycyrrhiza glabra* herb, its root and extracts.

**Figure 2 animals-09-00536-f002:**
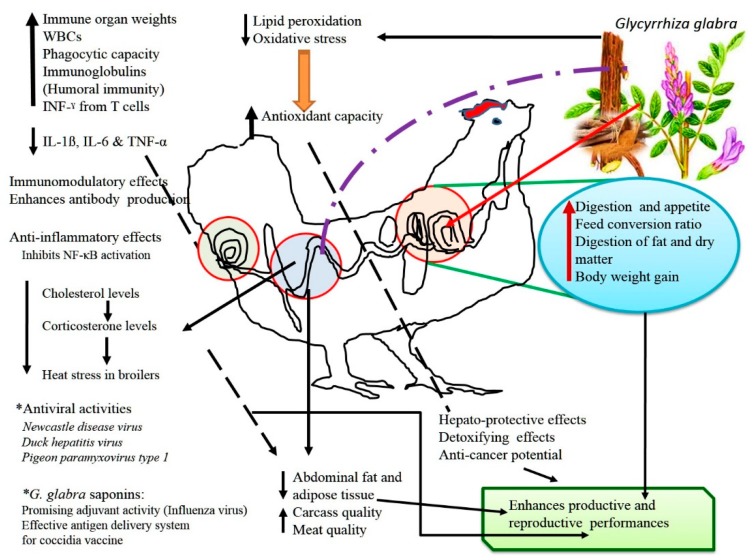
Beneficial effects and modes of action of *Glycyrrhiza glabra* in poultry health and production.

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
