# Peer review of "Use of Licorice (Glycyrrhiza glabra) Herb as a Feed Additive in Poultry: Current Knowledge and Prospects"

_animals, 2019, doi:10.3390/ani9080536_

Round 1
Reviewer 1 Report
Manuscript revision ID: animals-566993 titled:
Use of licorice (Glycyrrhiza glabra) herb as a feed additive in poultry: Current knowledge and prospects
The content of manuscript in chapters 4,5 and 6 should contain information pertaining to poultry experiments. The literature should be verified in terms of timeliness. Older than 10 years references are cited.
L137-149 – the information is not based on poultry research. Cited literature No: 12, 21, 29, 34-37 – does not apply to poultry.
L152-164 – there are no examples of research on poultry
L167-168 – what kind of health problems prevents licorice in poultry production?
L172-197 – Please, cite research on poultry. References No. 50, 51, 52, 54, 55, 56, 57, 58, 59 and 60 – does not apply to poultry.
Conclusions
Based on the analyzed literature, specify what forms and doses G.glabra should be administered for poultry to obtain beneficial health and production effect, please.
References
References No:17, 26, 62, 72, 85, 86, 87, 88, 90, 94, 95, 96, 101 – not current - older than 10 years.
References No: 23, 25, 30, 32, 36, 72, 107 – should be formatted in accordance with the Journal's requirements.
Author Response
Reviewer 1:
Moderate English changes required
English improved as suggested (edited by editors of www.editage.com, certificate was enclosedearlier)
Comments and Suggestions for Authors
Manuscript revision ID: animals-566993 titled:
Use of licorice (Glycyrrhiza glabra) herb as a feed additive in poultry: Current knowledge and prospects
The content of manuscript in chapters 4, 5 and 6 should contain information pertaining to poultry experiments. The literature should be verified in terms of timeliness. Older than 10 years references are cited.
Chapter 4 has been deleted and the other two chapters have been modified as suggested. Oder references deleted.
L137-149 – the information is not based on poultry research. Cited literature No: 12, 21, 29, 34-37 – does not apply to poultry.
Deleted and modified / revised as suggested.
References deleted and modified, Reference no. 29 and 36 retained as seem to be important. Reference 36 shifted in antiviral section.
L152-164 – there are no examples of research on poultry
Deleted and modified / revised as suggested.
L167-168 – what kind of health problems prevents licorice in poultry production?
This line is deleted.
L172-197 – Please, cite research on poultry. References No. 50, 51, 52, 54, 55, 56, 57, 58, 59 and 60 – does not apply to poultry.
References deleted and modified as suggested. Reference no. 29 and 36 retained as seem to be important. Reference 36 shifted in antiviral section.
Conclusions
Based on the analyzed literature, specify what forms and doses G. glabra should be administered for poultry to obtain beneficial health and production effect, please.
Doses mentioned.
References
References No: 17, 26, 62, 72, 85, 86, 87, 88, 90, 94, 95, 96, 101 – not current - older than 10 years.
References deleted as suggested.
References No: 23, 25, 30, 32, 36, 72, 107 – should be formatted in accordance with the Journal's requirements.
Formatted
The MS has also been updated with few recent references related to poultry on Licorice.
**Altogether 47 references have been deleted, and the MS updated with 06 most recent references related specifically to poultry and licorice.
*Only very few (2-3) references that are important with respect to Licorice herb have been kept / retained (which we could not delete as suggested pl..)
Reviewer 2 Report
L61: Refs 2, 3, and 4 are not supporting a statement associated with the "health of poultry." L63 still has ref 7 (L400) which does not support that statement. L64-65 I disagree that licorice is one of "the most extensively" used herbs in animal and poultry diets. Provide figure legends. Table 1: provide specific references corresponding to the individual components. Are all of the beneficial effects mentioned in Table 2 provided in the text? Please specify in your response to me so I can ensure accuracy. L303 reference 96 is still a mouse paper that the authors are suggesting shows a "positive influence" on "growth and laying performances of poultry" L305 an increase in FCR is not usually a benefit L333 "along with" what? L359 reviews are primary experiments that are testing hypothesisAuthor Response
English language and style are fine/minor spell check required
English language corrected.
Comments and Suggestions for Authors
L61: Refs 2, 3, and 4 are not supporting a statement associated with the "health of poultry."
Refs 3 & 4 deleted, however Refs 2 – Dhama et al., 2018 - supports health of poultry, as stated in abstract and details in Section 5.6 of this MS published.
L63 still has ref 7 (L400) which does not support that statement.
Ref 7 deleted
L64-65 I disagree that licorice is one of "the most extensively" used herbs in animal and poultry diets
Deleted
Provide figure legends.
Figure legends are provided.
Table 1: provide specific references corresponding to the individual components. Are all of the beneficial effects mentioned in Table 2 provided in the text? Please specify in your response to me so I can ensure accuracy.
Table 1 has been deleted, as sufficient and relevant information was present in text. It was a general table, so deleted now.
L303 reference 96 is still a mouse paper that the authors are suggesting shows a "positive influence" on "growth and laying performances of poultry"
Appropriate / relevant reference added.
L305 an increase in FCR is not usually a benefit
It is deleted
L333 "along with" what?
Corrected.
L359 reviews are primary experiments that are testing hypothesis.
Modified as suggested.
**Altogether 47 references have been deleted, and the MS updated with 06 most recent references related specifically to poultry and licorice.
The MS has also been updated with few recent references related to poultry on Licorice.
This manuscript is a resubmission of an earlier submission. The following is a list of the peer review reports and author responses from that submission.
Round 1
Reviewer 1 Report
please find review opinion in attachment

Author Response
1. Please change the tittle: “Use of Licorice (glycyrrhiza glabra) herb as feed additive in poultry – Current Knowledge and Prospects”
Replaced
2. Delete the sentence “Hence, it will be highly useful for nutritionists, physiologists, pharmacists, veterinarians, and poultry producers” in line 52-53
Deleted
3. In line 59: “….in animals and humans”. delete “, and their importance is realized over the world”
Deleted
4. In line 61: Did “growth promoters” mean “antibiotic”? However, antibiotics do not promoter growth directly. Meanwhile, please replace the word “nutraceuticals” “herbs” with “prebiotic” “plant extracts”
Corrected
5. The format of the tables is incorrect. Please follow the author guideline of the journal
Corrected
6. L137: change the sub-tittle “Beneficial role of licorice”
Changed
7. In line 410: please rewrite this sentence. Did you mean “245 one-day-ole broiler chickens”?
Corrected
8. In line 430, use “….play an important role in….”
Replaced
9. In line 432, use “contains bioactive components such as…”; “, which possess pharmacological activities….”
Corrected
10. Delete line 433-434: “The major advantage of …….without any deleterious effects.”
Deleted
11. Delete line 435-438, use “The licorice has found to show immunogenic and antioxidant activity, which may improve the growth performance, feed efficiency, carcass traits, blood biochemical indices, and be a potential solution for solving the respiratory, digestive and immune problem in poultry.”
Done accordingly
12. Delete line 439-443, use “Further researches need to be conducted to evaluate the consistence of the beneficial effects of using licorice herb as feed additives in poultry, as well as to explore other properties of this medicinal plant that may enhance productivity and health in poultry.”
Deleted and done accordingly
13. Please follow the author guideline to change the format of the References section
Done accordingly
The revised MS as been also updated relevantly.
Reviewer 2 Report
Manuscript revision ID: animals-518178
Licorice (glycyrrhiza glabra) herb as an eco-friendly additive to promote poultry health – Current Knowledge and Prospects
The manuscript can’t be published as such form in Animals. The Authors have taken up a very interesting topic. The manuscript content in large part doesn’t correspond to the title. It contains information about research on laboratory animals or humans - used in human medicine. This part of manusctipt does’nt take into account the poultry physiology and poultry production specific. Manuscript does’nt present the "Current Knowledge" - is outdated literature cited, and there is no "Prospects" - no description of licorce wide introduction for poultry production. It should be thoroughly rewritten and adapted to the title and specifics of the journal, or change the type of journal.
Abstract
Please, limit only to information on poultry (according to the title). An application summary is missing - how the form and what amount of licorice can be recommended in poultry production.
Introduction
L79-84 – information about children and Drosophila melanogaster, not poultry.
L85-93 – to what extent is this information useful in poultry production. In the animal husbandry is important to achieve rapid weight gain of birds ... - not the weight loss….
L94-98 - the aim of the study is formulated too generally. It does not correspond to the expectations of the poultry production sector
L103-107 – Please provide detailed information on the chemical composition and amount of individual biologically active substances in liquorice. Which of these substances are present in the dominant quantities?
L108-118 - information inadequate to the subchapter - they do not concern the chemical structure.
L138-142 - lack of literature and references to reliable poultry studies
L148-149 - information on the history of treatment unnecessary, please delete
L159-177 - description of the licorice therapeutic effect does not apply to poultry! It's mainly human medicine.
L170 - what is the diagnostic and therapeutic treatment of Addison's disease in birds? The authors write about the effects of its treatment in pediatrics !! I think they chose the magazine wrong.
L194-222 - research has nothing to do with the production and breeding of poultry.
L264-288 - this is the first paragraph corresponding to the manuscript title.
It is important to verify the cited studies. The authors can not mix the results of research conducted on farm animals and people - they are not the same.
Conclusions
L430-433 - Too general. Application indications are missing.
Table 1 – add sources of literature, please. What are the amounts of these substances in root lichenice.
Table 2 - studies on birds are not presented, only on laboratory animals in the aspect of human medicine, not veterinary medicine
References
This is a review of knowledge - it should be current. Please, cite studies no older than 5-10 years. In the manuscript are quoted about 38% of studies older than 10 years and 8.5% - up to 36 years old !!! This is not "Current Knowledge".
Author Response
The manuscript can’t be published as such form in Animals. The Authors have taken up a very interesting topic. The manuscript content in large part doesn’t correspond to the title. It contains information about research on laboratory animals or humans - used in human medicine. This part of manusctipt doesn’t take into account the poultry physiology and poultry production specific. Manuscript doesn’t present the "Current Knowledge" - is outdated literature cited, and there is no "Prospects" - no description of licorce wide introduction for poultry production. It should be thoroughly rewritten and adapted to the title and specifics of the journal, or change the type of journal.
Abstract
Please, limit only to information on poultry (according to the title). An application summary is missing - how the form and what amount of licorice can be recommended in poultry production.
Done accordingly
Introduction
L79-84 – information about children and Drosophila melanogaster, not poultry.
All information concerned animals and human and not belong to poultry were deleted from the review
L85-93 – to what extent is this information useful in poultry production. In the animal husbandry is important to achieve rapid weight gain of birds ... - not the weight loss….
Corrected
L94-98 - the aim of the study is formulated too generally. It does not correspond to the expectations of the poultry production sector
Done accordingly
L103-107 – Please provide detailed information on the chemical composition and amount of individual biologically active substances in liquorice. Which of these substances are present in the dominant quantities?
Required information added
L108-118 - information inadequate to the subchapter - they do not concern the chemical structure.
Done accordingly
L138-142 - lack of literature and references to reliable poultry studies
Required information added
L148-149 - information on the history of treatment unnecessary, please delete
Deleted
L159-177 - description of the licorice therapeutic effect does not apply to poultry! It's mainly human medicine.
All information on other animals and human than poultry were deleted from the review
L170 - what is the diagnostic and therapeutic treatment of Addison's disease in birds? The authors write about the effects of its treatment in pediatrics !! I think they chose the magazine wrong.
Deleted and corrected
L194-222 - research has nothing to do with the production and breeding of poultry.
Several improvements have been done in the revised paper.
L264-288 - this is the first paragraph corresponding to the manuscript title.
Thanks for your comment
It is important to verify the cited studies. The authors can not mix the results of research conducted on farm animals and people - they are not the same.
All information concerned animals and human and not belong to poultry were deleted from the review
Conclusions
L430-433 - Too general. Application indications are missing.
Done accordingly as a response to comments of the first reviewer advised some sentences
Table 1 – add sources of literature, please. What are the amounts of these substances in root lichenice.
Data in Table 1 such as Chemical formula, Molecular Weight, Chemical Names and PubChem CID were collected from PubChem (a database of chemical molecules), For examples:
Glycyrrhizin: Please look https://pubchem.ncbi.nlm.nih.gov/compound/Glycyrrhizin
Dehydroglyasperin C: Please look https://pubchem.ncbi.nlm.nih.gov/compound/Dehydroglyasperin_C
Licoricidin: Please look https://pubchem.ncbi.nlm.nih.gov/compound/Licoricidin
In addition to, the amounts of these substances in licorice root vary depending on plant origin, plant maturity, extraction method and many other factors ... etc.However, cultivation and harvesting practices modify the composition of various biologically important components of Glycyrrhiza plant. So it was difficult tocollect the quantity of each substance in this manuscript
Table 2 - studies on birds are not presented, only on laboratory animals in the aspect of human medicine, not veterinary medicine.
Deleted
References
This is a review of knowledge - it should be current. Please, cite studies no older than 5-10 years. In the manuscript are quoted about 38% of studies older than 10 years and 8.5% - up to 36 years old !!! This is not "Current Knowledge".
The revised MS as been also updated relevantly.
Thank you very much for give us a chance to improve our paper. NOW, we updated the paper based on your suggestions. Thanks again for your efforts .
Reviewer 3 Report
There are many references throughout the text that do not directly support your statements. Some examples include #7 (primary zoster paper not a historical use of licorice paper), #11 (extraction process paper not supporting the long list of activities listed), #109 (rotavirus paper), #131 (mouse kidney disease paper not a chicken paper supporting the use of feed additives).
In addition to the mis-representation of the above references (and others), the authors make statements suggesting the findings are related to poultry when in fact they are not. For example, L138-142 state health benefits associated with licorice use in mammals (cough medicine, hepatitis, mouth ulcers, etc.) then Figure 2 is chicken-specific. Basically, it seems the authors use mammalian studies and indicate the same is true for poultry (L159-177). In other instances, no references are provided to support your statements. Another example is L195-202 (no reference is given for the poultry statement then the authors use examples from the mammalian literature without clarification).
What is the relevance of a paragraph dedicated to Hep C?
Author Response
There are many references throughout the text that do not directly support your statements. Some examples include #7 (primary zoster paper not a historical use of licorice paper), #11 (extraction process paper not supporting the long list of activities listed), #109 (rotavirus paper), #131 (mouse kidney disease paper not a chicken paper supporting the use of feed additives).
Deleted and done accordingly
In addition to the mis-representation of the above references (and others), the authors make statements suggesting the findings are related to poultry when in fact they are not. For example, L138-142 state health benefits associated with licorice use in mammals (cough medicine, hepatitis, mouth ulcers, etc.) then Figure 2 is chicken-specific. Basically, it seems the authors use mammalian studies and indicate the same is true for poultry (L159-177). In other instances, no references are provided to support your statements. Another example is L195-202 (no reference is given for the poultry statement then the authors use examples from the mammalian literature without clarification).
All information concerned animals and human and not belong to poultry were deleted from the review
What is the relevance of a paragraph dedicated to Hep C?
All information concerned animals and human and not belong to poultry were deleted from the review
The revised MS as been also updated relevantly.